# Comparative Mapping of the Macrochromosomes of Eight Avian Species Provides Further Insight into Their Phylogenetic Relationships and Avian Karyotype Evolution

**DOI:** 10.3390/cells10020362

**Published:** 2021-02-09

**Authors:** Lucas G. Kiazim, Rebecca E. O’Connor, Denis M. Larkin, Michael N. Romanov, Valery G. Narushin, Evgeni A. Brazhnik, Darren K. Griffin

**Affiliations:** 1School of Biosciences, University of Kent, Canterbury CT2 7NJ, UK; lkiazim@gmail.com (L.G.K.); rebeckyoc@gmail.com (R.E.O.); M.Romanov@kent.ac.uk (M.N.R.); 2Department of Comparative Biomedical Sciences, Royal Veterinary College, University of London, London NW1 0T, UK; dlarkin@rvc.ac.uk; 3Research Institute for Environment Treatment, 69032 Zaporozhye, Ukraine; val@vitamarket.com.ua; 4Vita-Market Ltd., 69032 Zaporozhye, Ukraine; 5BIOTROF+ Ltd., Pushkin, St. Petersburg 196602, Russia; bea@biotrof.ru

**Keywords:** avian species, macrochromosome, comparative cytogenetic maps, BACs, chromosome rearrangements, phylogenomics

## Abstract

Avian genomes typically consist of ~10 pairs of macro- and ~30 pairs of microchromosomes. While inter-chromosomally, a pattern emerges of very little change (with notable exceptions) throughout evolution, intrachromosomal changes remain relatively poorly studied. To rectify this, here we use a pan-avian universally hybridising set of 74 chicken bacterial artificial chromosome (BAC) probes on the macrochromosomes of eight bird species: common blackbird, Atlantic canary, Eurasian woodcock, helmeted guinea fowl, houbara bustard, mallard duck, and rock dove. A combination of molecular cytogenetic, bioinformatics, and mathematical analyses allowed the building of comparative cytogenetic maps, reconstruction of a putative Neognathae ancestor, and assessment of chromosome rearrangement patterns and phylogenetic relationships in the studied neognath lineages. We observe that, as with our previous studies, chicken appears to have the karyotype most similar to the ancestor; however, previous reports of an increased rate of intrachromosomal change in Passeriformes (songbirds) appear not to be the case in our dataset. The use of this universally hybridizing probe set is applicable not only for the re-tracing of avian karyotype evolution but, potentially, for reconstructing genome assemblies.

## 1. Introduction

Most birds exhibit a highly distinctive, “typical” avian karyotype, where chromosomes are characteristically divided into around 10 macrochromosomes and around 30 similarly-sized, morphologically indistinguishable microchromosomes. Around two thirds of species have a karyotype 2n = 76–82 [1,2,3]. Studying overall genome structure is an essential element to understanding avian biology; however, most avian species have no structural (karyotypic) data associated with their genome [4] despite ~460 avian genomes having been sequenced (4% of all species). As of May 2019 [5], only 16 genomes have been assembled to a chromosome-level (i.e., a single scaffold for each chromosome from the *p*- to *q*-terminus). To address this problem coupling classical cytogenetics with molecular cytogenetics, such as fluorescence in situ hybridisation (FISH), provides a finer resolution of genomic structure and can be used to anchor genome sequence to the chromosomes and thence identify chromosome rearrangements by determining interspecies homology. Chromosome painting by FISH has resulted in numerous comparative genomic and evolutionary studies in birds (e.g., [3,6,7,8,9,10,11,12]); however, chromosome paints are limited in their ability to identify intrachromosomal rearrangements such as inversions and duplications. These limitations can be circumvented through the use of bacterial artificial chromosome (BAC) clone probes, providing a finer resolution to detect small rearrangements. Through the use of a universal BAC probe set developed by Damas et al. [13], these rearrangements can be mapped by measuring the fractional length relative to the p-terminus (FLpter) value [14]. Using a reference genome with known BAC order for comparison, the mapping of BACs can thereafter be used to track chromosomal rearrangements between different species, providing an inexpensive way to characterise genomic rearrangements without the need for sequencing data. These data can also be used to generate comparative maps that lay the foundation for other studies, including upgrading assemblies to a chromosome-level [13,15].

In order to explore the extent of chromosomal rearrangement in the macrochromosomes of birds relative to chromosomes of the chicken (*Gallus gallus*; the order Galliformes), the best cytogenomically studied bird [16,17,18], and ultimately, a reconstructed hypothetical ancestor for the infraclass Neognathae, we selected seven further avian species, providing representatives for six of the 32 neognath orders (Table 1). These included two songbirds, the common blackbird (*Turdus merula*, TME), and Atlantic canary (*Serinus canaria*, SCD), both representatives of the order Passeriformes. Passerine birds comprise over half of all avian species [19]; known for their phenotypic diversity and for their vocal learning, they are often used for studies related to brain development [20,21]. An assembled and annotated canary genome is available; however, not currently to a chromosome-level [22]. Further selected species included the Eurasian woodcock (*Scolopax rusticola*, SRU; Charadriiformes), houbara bustard (*Chlamydotis undulata*, CUN; Otidiformes), and the rock dove, or pigeon (*Columba livia*, CLI; Columbiformes). The Eurasian woodcock is a wading bird known for its 360-degree vision and recognised among game hunters for its erratic flight patterns, speed, and size in addition to having an atypical diploid number of 2n = 96 [23,24,25]. The houbara bustard is culturally significant in Arabian countries, in addition to being listed by the IUCN as vulnerable [26]. The pigeon exhibits extreme phenotypic diversity not seen within any wild avian species [27]. The pigeon genome has also recently been upgraded to a chromosome-level [13], thereby providing an additional reference point for BAC mapping in other species and ensuring further validation of this method for identifying chromosomal rearrangements without sequencing data. Among the basal superorder Galloanserae, the mallard duck (*Anas platyrhynchos*, APL; Anseriformes) and the helmeted guinea fowl (*Numida meleagris*, NME; Galliformes) were chosen; the former being a particularly well explored species for immunology studies [28] as well as having a whole genome radiation hybrid panel [29]. It also last shared a common ancestor with chicken more recently than the other species in this study (~47 million years ago; [30]). Finally, the helmeted guinea fowl provides an additional representative of the Galliformes, ensuring that any chicken specific features are not overly represented for this order in this dataset.

In these species, we used a pan-avian universally hybridising probe set [13,15,31] to trace macrochromosomal evolution through the generation of comparative cytogenetic maps. This approach permits identification of fissions, fusions, duplications, and inversions, all of which contribute to the chromosomal changes that influence speciation, the phylogenetic relationships between the eight species and the lineage-specific chromosomal rearrangements as additionally explored with bioinformatic/mathematical tools.

## 2. Materials and Methods

### 2.1. Cell Culture and Chromosome Preparation

Chicken embryonic fibroblasts were obtained from The Pirbright Institute, Woking, Surrey, GU24 0NF, UK (chicken embryonic fibroblast DF-1). All other cell lines for the common blackbird, Atlantic canary, Eurasian woodcock, helmeted guinea fowl, houbara bustard, mallard duck, and rock dove (pigeon) are available through the Malcolm Ferguson-Smith collection, CryoArks, National History Museum, London, UK, and are available on request via enquiries@cryoarks.org (there are no specific accession numbers). No animals were used in the course of this study and hence no ethical oversight required. Briefly, fibroblasts were cultured at 40 °C with 5% CO_2_ in Alpha MEM (Gibco/Thermo Fisher Scientific, Inc., Waltham, MA, USA), supplemented with 10% foetal bovine serum (Gibco) and 1% Penicillin-Streptomycin-L-Glutamine (Sigma-Aldrich, St. Louis, MO, USA). Chromosome suspension preparation followed standard protocols where colcemid solution (Gibco) at a concentration of 5.0 μg/mL was added to flasks for 1 h prior to hypotonic treatment with 75 mM pre-warmed (37 °C) KCl for 1 h and subsequent fixation in 3:1 absolute methanol:glacial acetic acid.

### 2.2. Preparation of BAC Probes for FISH

The 74 chicken BACs were selected as a pan-avian universally hybridising probe set according to Damas et al. [13] and O’Connor et al. [15,31,32]. DNA was isolated from BAC clones using the QIAGEN (Hilden, Germany) miniprep kit and was subsequently amplified, then labelled directly by nick translation. Probes were labelled with Texas red-12-dUTP (Invitrogen/Thermo Fisher Scientific, Inc., Waltham, MA, USA) and FITC-fluorescein-12-UTP (Roche Diagnostics, Rotkreuz, Switzerland) prior to purification with the QIAGEN nucleotide removal kit.

### 2.3. Fluorescence In Situ Hybridisation (FISH)

Metaphase preparations were fixed to slides and run through an ethanol series (2 min each in 2× SSC, 70%, 85%, and 100% ethanol at room temperature). Dual colour FISH was set up with FITC and Texas Red labelled probes mixed with COT-1 DNA (Insight Biotechnology, Wembley, UK) and Hybridisation solution I (Cytocell Ltd., Cambridge, UK). Probe and target DNA were simultaneously denatured on a 75 °C hotplate for 2 min, then left to hybridise in a humidified chamber for 72 h at 37 °C. Post-hybridisation washes were 30 s in 2× SSC/0.05% Tween 20 at room temperature prior to counterstaining using VECTASHIELD antifade mounting medium with DAPI (Vector Laboratories, Inc., Burlingame, CA, USA). Slides were visualised under an Olympus BX61 epifluorescence microscope. A cooled CCD camera captured images with DAPI, Texas Red, FITC, and Aqua filters. Images were captured at 100× magnification using SmartCapture3 software (Digital Scientific UK, Cambridge, UK).

### 2.4. Karyotype Analysis and Ideogram Generation

Taking into consideration the nomenclature described by the International System for Standardised Avian Karyotypes [33] regarding chromosome size, karyotype images were produced per species using SmartType3 software (Digital Scientific, UK). In the case of the songbirds, the nomenclature describing chicken chromosome homology was used. Ideograms were created based on the karyotype images produced using Microsoft PowerPoint (Microsoft, Redmond, WA, USA). Banding patterns were replicated by visual interpretation, with measurements (where possible) being made for a degree of accuracy. The results were verified by comparing multiple karyotype images to account for any variance in banding between metaphase spreads. FLpter measurements were made for each probe using ImageJ [34]. For visual clarity, the 74 BACs were numbered in ascending order based on their position on the chicken chromosome, with number 1 being at the topmost position of the p-arm. Using the species-specific ideograms, the comparative cytogenetic maps were generated showing the position for 74 BAC hybridisation sites and centromeres on each chromosome.

### 2.5. MLGO (Maximum Likelihood for Gene Order Analysis) Analysis for Ancestral Genome Reconstruction

To deconvolute lineage-specific rearrangement patterns in the seven new species plus the reference chicken genome, we first reconstructed in situ chromosomes for their most common ancestor, i.e., hypothetical ancestor for all Neognathae birds. An estimation of the neognath ancestral genome was inferred using the MLGO (Maximum Likelihood for Gene Order Analysis) web server [35]. The advantage of this reconstruction tool is that it can handle not only simple rearrangements like inversions, but also insertion, deletion, duplication and translocation events, while being capable to process large-scale datasets for nuclear genomes including information of missing BACs if any of them failed to hybridise in any species. To reconstruct the presumptive neognath ancestral genome, we employed the FISH data for the ostrich (*Struthio camelus*; order Struthioniformes, infraclass Palaeognathae) as an outgroup that were obtained in our previous study [15] using the same pan-avian 74-BAC set [13,15,31,32]. As an MLGO input phylogenetic tree for the eight birds plus ostrich, we took as a basis the tree shown in Figure 1 that was derived from the comprehensive [36] phylogeny for the class Aves.

Importantly, using MLGO, we were also able not only to treat chromosome-specific order of the 74 BAC hybridisation sites, but also introduce the individually numbered centromeres. This enabled to compose species-specific MLGO input datasets for up to 84 sites per genome, with the position of the 74 BAC hybridisation sites being oriented relative to each other and to the respective centromere. This approach provided the most appropriate reconstruction of the suppositive neognath ancestral genome as well as the most accurate and precise assessment of the possible lineage-specific intra- and interchromosomal rearrangements in the eight birds studied.

### 2.6. GRIMM (Genome Rearrangements In Man and Mouse) Analysis for Chromosome Rearrangement

Using the MLGO-assisted reconstructed Neognathae ancestral genome, we further exploited the GRIMM (Genome Rearrangements in Man and Mouse) web tool [38] to examine lineage-specific rearrangement patterns among the infraclass Neognathae. GRIMM analysis involved design of multichromosomal genome inputs based on the same chromosome-specific order of the 74 BAC hybridisation sites in the putative neognath ancestor and eight birds studied using the same format for a genome representation as in the MLGO datasets. Intra- and interchromosomal rearrangements were explored and summarised in pairs of genomes, with one being the neognath ancestor and the other one being an individual bird. The pairwise GRIMM outputs were double checked manually to ensure the correct rearrangement assignment and interpretation.

### 2.7. Mathematical Analyses

To estimate relations between lineage-specific rearrangement profiles and genome divergence among the eight studied bird karyotypes, the respective graphs were built, and pairwise Pearson’s/multiple correlation coefficients were calculated using Microsoft Excel and STATISTICA 5.5 (StatSoft, Inc./TIBCO, Palo Alto, CA, USA; see details in Appendix A). Principal Component Analysis (PCA), Fuzzy Analysis Clustering (FAC), and average linkage clustering (ALC) were performed using R and libraries for R environment, and Euclidean distance metric (see details in Appendix A).

## 3. Results

### 3.1. Karyotypes and Ideograms for Eight Avian Species

Conventional analysis of metaphases from all eight avian species revealed diploid numbers ranging between 76 and 96 chromosomes. Table 1 summarises the chromosomal findings of each species studied.

Karyotypes were completed based on existing studies [10,13,15,32,40,41,42]. The houbara bustard had conflicting karyotype data in the literature, with either a diploid number of 76 [32] or 78 [43]. However, karyotypes performed for this study determined a diploid number of 76. For species where no literature was present, karyotypes were completed following ISSAK classifications [33]. An example of the typical avian karyotype (2n = ~80) is shown in Figure 2A,B, representing the chicken and the mallard duck. Figure 2C,H, representing the Eurasian woodcock and the Atlantic canary, demonstrate different karyotypes that vary either in diploid number or deviate from the ISSAK classification of being ordered by size (chromosome 1 and 1A being ordered before chromosome 2, the largest chromosome in the Atlantic canary).

Using visual inspection and measurements of the chromosome arms, respective chromosome length, and width of bands, ideograms were generated from the karyotypes of the macrochromosomes (1–9, Z, and W). These ideograms (shown in Figure 3) demonstrate differences in chromosome morphology and banding. For instance, helmeted guinea fowl chromosomes in Figure 3E are more heavily banded than chicken chromosomes in 2A, which may not have been apparent in the karyotype images.

### 3.2. Application of a Panel of 74 Selected Chicken BACs for the Fine Mapping of Macrochromosome Homologs 1–9 and Z

The 74 conserved BAC clones were selected based on work developed by Damas et al. [13] for hybridisation to the macrochromosomes, with the complete list of BACs and their coordinates in the chicken genome given in the Appendix A. The degree of successful hybridisations varied between species, with an overall success rate for all 74 BACs given in Table 1.

For BACs that were successfully hybridised (as exemplified in Figure 4 and Figure 5), FLpter values, standard deviations, and the number of mitotic chromosomes measured were recorded. The full tables of results for all species are shown in Appendix A.

### 3.3. Reconstructing the Neognathae Ancestor and Rearrangements

Following MLGO and GRIMM analyses (see Materials and Methods) and using the eight species in this study and ostrich as an outgroup, the presumed neognath ancestral karyotype was reconstructed and the number of changes that occurred from the ancestor noted (Table 2). Chicken had the lowest number (4) followed by guinea fowl (6), duck (8), houbara (9), blackbird (10), canary and pigeon (11 each), and woodcock with the most at 16. The 11 canary rearrangements included two apparent duplications in which extra BAC signals were clearly seen in this species but not in others (and not for other BACs). Intrachromosomal rearrangements identified were both inversions and intrachromosomal duplications. The interchromosomal rearrangements consisted of fusions, fissions, interchromosomal duplications, and translocations. The greater number seen in woodcock is largely accounted for by inter-chromosomal rearrangements (including fissions) whereas pigeon had the most intra-chromosomal rearrangements.

Correlation analysis for the eight avian species (see Appendix A) revealed an association between chromosomal rearrangement patterns, on the one hand, and overall karyotype/genome organisation and divergence time, on the other. PCA/FAC/ALC-based assessments (see Appendix A) provided further information in support of the cytogenetically observed specifics of chromosome changes within and between individual lineages in the infraclass Neognathae.

In particular, there was a certain linear correlation between the percentage of failed chicken BAC probes and total numbers of rearrangements in the eight birds (*R*^2^ = 0.6378, *p* < 0.05; Appendix A). When we used an integrative genome/divergence index (IGDI; see details in Appendix A), this showed a higher linear correlation with total number of rearrangements (*R*^2^ = 0.8427, *p* < 0.01; Figure 6A).

By employing the multiple correlation procedure (see details in Appendix A), a 3D surface plot was produced (Figure 6b) that demonstrates that total number of rearrangements (VAR1) grows if both FISH success rate (VAR2) and ratio of diploid number of chromosomes to the typical avian karyotype (2n/80; VAR3) increase (*R* = 0.983, *p* < 0.001).

The eight-species clustering patterns in the correlation and PCA/FAC/ALC analyses (see details in Appendix A) were in agreement with the known phylogeny for this set of the eight neognath birds (Figure 1), except for the duck tending to be closer to the houbara–pigeon pair than to the chicken–guinea fowl pair (see a clustering example on the PCA score plot in Figure 7).

## 4. Discussion

Use of a universally hybridizing BAC set was successful in detecting multiple chromosomal rearrangements during evolution of the eight species studied. Where our previous studies have used this approach to determine that microchromosomes have undergone few chromosomal rearrangements throughout evolution (O’Connor et al., 2019), the results presented here show that macrochromosomes exhibit both intra- and interchromosomal rearrangements, with the type and number of rearrangement dependent on the lineage. Compared to mammals however, changes are still relatively few, although rearrangement rates can be variable in different lineages (e.g., [44]). A number of patterns emerge, first that, in agreement, with several of our previous studies, chicken appears to be the genome organisation closest to the ancestor. Second however, while previous studies have shown a greater number of intrachromosomal rearrangements in Passeriformes, we find no such evidence, with pigeon, woodcock, and houbara all having similar numbers to canary and blackbird. Previous studies have correlated the greater number of intrachromosomal rearrangements with greater levels of speciation in songbirds (e.g., [45]). This hypothesis may need to be re-assessed in the light of these results.

### 4.1. Comparative Macrochromosome Maps

Comparative mapping provides insight into patterns of conservation and rearrangement between species. For some chromosomes, there were rearrangement patterns already observed between species within the same order (chromosome 1 fission in the common blackbird and Atlantic canary), which were absent when compared to species from other orders. Other examples can be seen in chromosome 5 for the blackbird, canary, and woodcock, and chromosome 7 in the blackbird, guinea fowl, houbara, and pigeon; each of these patterns is usually in the form of an inversion of the same loci.

The use of comparative mapping aids in the identification of homologous synteny blocks and the evolutionary breakpoints between them, both of which contribute to the evolutionary changes that result in lineage-specific traits (e.g., [31]). However, it is widely debated whether patterns of chromosome evolution are caused by fixed deleterious mutations or high mutation rates resulting in genetic drift [46,47,48]. Nevertheless, chromosomal rearrangements have been found to play a role in speciation as a result of enhanced reproductive isolation through reduced hybrid fitness, and also due to barriers to gene flow in non-recombining regions [49,50].

Moreover, the identification of patterns between species despite divergence times of millions of years signifies an evolutionary role in promoting speciation. For example, the inversions identified here indicate the occurrence of double stranded DNA breaks, and the recurrent use of these breakpoints are due to fragile genomic regions [51]. Larkin et al. [52] established that these evolutionary breakpoint regions have a propensity for promoting chromosomal rearrangement as they are found within gene-dense areas, in which the genes are related to lineage-specific traits [53,54,55]. We later demonstrated that genes near evolutionary breakpoints have a higher chance to change expression profiles due to regulation modifications caused by novel enhancers [56]. Furthermore, it can be suggested that the recurrent breakpoint use could generate novel combinations of genes/regulation networks that may help to promote adaptation.

### 4.2. Chromosome Paints vs. BAC Mapping

The generation of avian chromosome paints [57] was a significant breakthrough for comparative studies, allowing for the detection of large syntenic relationships between both closely and distantly related species. These chromosome paints have been tested on more than 70 different species (e.g., [3,6,9,58,59]). However, there are many limitations with chromosome paints that restrict comparative studies: the orientation of syntenic regions cannot be established, meaning any number of inversions could be undetected. Moreover, cross-species chromosome painting can yield ambiguous results with non-specific binding, which could either be interpreted as a duplication or translocation, or if a small rearrangement is present, it could be dismissed entirely.

Some of the species studied here have previously had chromosome paints applied to their macrochromosomes [60,61,62], with the main conclusion being that there was high conservation of synteny. Whilst fissions and fusions were detected, the depth of detail provided by the paints was limited. The availability of avian genomic sequences for a well-defined library of BACs has increased the number of genetic markers, allowing for a greater detection of chromosomal rearrangements. For example, studies of the helmeted guinea fowl have shown a fusion of chromosome 6 and 7 to form chromosome 5 (when ordered by size). The BAC mapping in this study not only detected this fusion, but also detected whether there were any intrachromosomal rearrangements within chromosomes 6 and 7, and which orientation the chromosomes fused. Thus, the resolution of detail provided in this study surpasses that of the chromosome painting data and provides more depth to comparative studies of avian species.

### 4.3. Centromere Position

An interesting aspect of these analyses is centromere position. There are several examples of the order of BACs not changing, but the relative position of the centromere being different compared to them. The “floating centromere” hypothesis was first proposed by Jackson [63,64], while “centromere repositioning” was more recently described in mammalian genomes (e.g., [65]). According to these observations, centromeres can disappear and re-form in different places on the chromosome, without changing gene order; the results presented here provide further evidence of this phenomenon in birds.

### 4.4. A Potential Tool for Genome Assembly

Finally, while some of the species studied here have chromosome-level genome assemblies associated with them, others do not. Indeed, some have not been sequenced at all. In previous studies, we designed BACs to help complete genome assemblies [13,15]. Here, we use the same BAC set on multiple species, demonstrating proof of principle that the data could be retrofitted to a genome assembly of sufficient quality, i.e., with sufficiently few large super-scaffolds such that, for the most part, at least one BAC will be located on them. While genome assemblies continue to improve, some achieving near-chromosome level, a BAC set such as this one would provide confirmatory evidence of the overall genome organisation (e.g., [66,67]).

### 4.5. Phylogenetics

In terms of phylogenetic relationships as revealed by correlation and PCA/FAC/ALC analyses (see details in Appendix A [68,69,70,71,72,73,74,75,76]), the observed species grouping basically followed the known phylogeny for this set of eight birds (Figure 1). The only exception was the duck that tends to be closer to the houbara–pigeon pair than to the chicken–guinea fowl pair that might be an evidence that resolution power of the selected 74 BAC probes for interspecies hybridisation, bioinformatic tools used and/or few additional divergence/karyotype characteristics chosen for mathematical analyses [68,69,70,71,72,73,74,75,76] seemed insufficient in verifying the known avian phylogeny for the duck.

## 5. Conclusions

In conclusion, the results presented here provide hitherto implicit information on the overall chromosome-level organisation of the avian genome and the changes that occurred from the suppositional common Neognathae ancestor. A universally hybridising BAC set is presented that is a main component of the toolbox useful both for genome assembly and phylogenomics of many avian (and possibly other reptilian) species. Comparative cytogenetic maps and chromosome changes analysed here for the eight birds exemplifies efficient applicability of cytogenomic techniques to tackle common and peculiar features of genome organisation and evolution in the class Aves [67]. The observed genomic “variadicity” and specific chromosomal rearrangements are compatible with the available information on general makeup, stability, and variation of the genomes in certain avian taxa with the reference to their ancestors [16,67].

## Figures and Tables

**Figure 1 cells-10-00362-f001:**
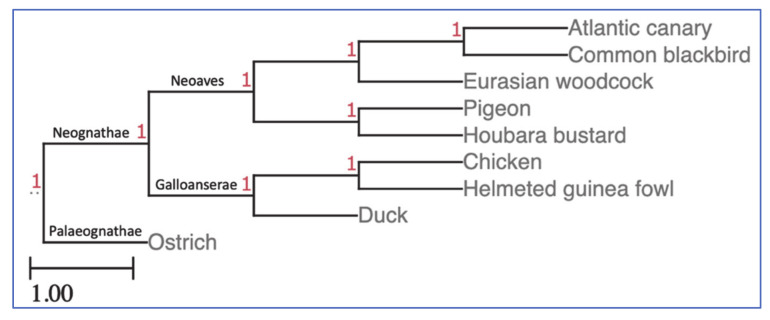
The Maximum Likelihood for Gene Order Analysis (MLGO input phylogenetic tree for the eight birds plus ostrich taken as an outgroup genome. The tree was visualised using the ETE v3 toolkit [37]. The respective Newick format tree can be written as (((((Atlantic canary, Common blackbird), Eurasian woodcock), (Pigeon, Houbara bustard)), ((Chicken, Helmeted guinea fowl), Duck)), Ostrich); as inferred from the Prum et al. [36] phylogeny for birds. Provisional support values (1) are shown in red.

**Figure 2 cells-10-00362-f002:**
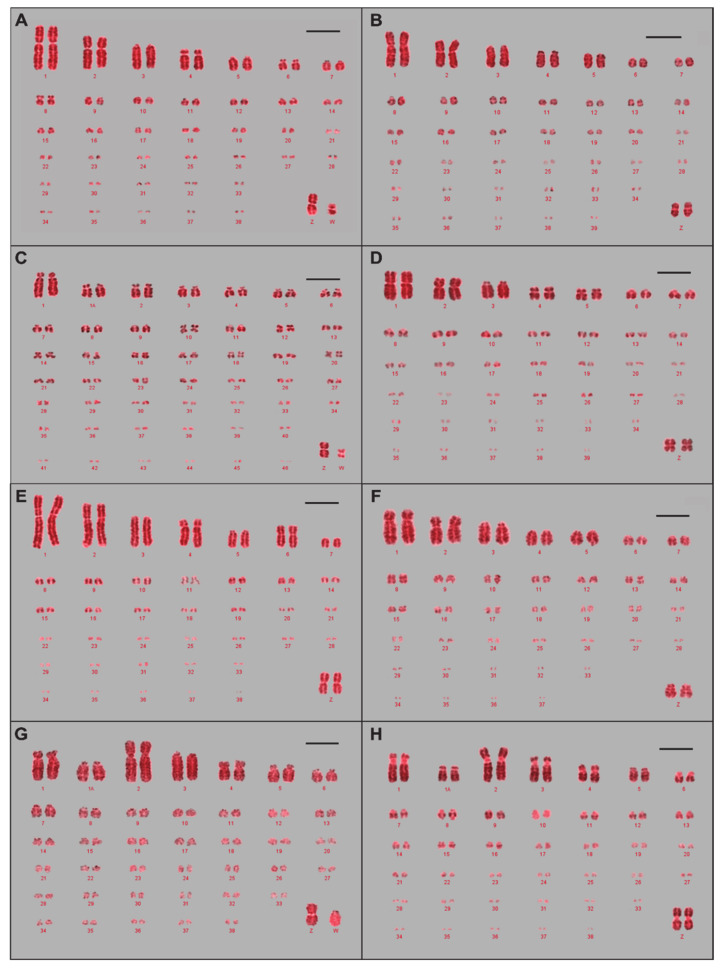
The variety of avian karyotypes observed in the seven avian species compared to chicken: (**A**) chicken (*Gallus gallus*), (**B**) mallard duck (*Anas platyrhynchos*), (**C**) Eurasian woodcock (*Scolopax rusticola*), (**D**) rock dove/pigeon (*Columba livia),* (**E**) helmeted guinea fowl (*Numida meleagris*), (**F**) houbara bustard (*Chlamydotis undulata),* (**G**) common blackbird (*Turdus merula*), (**H**) Atlantic canary (*Serinus canaria*). Scale bar 5 μm.

**Figure 3 cells-10-00362-f003:**
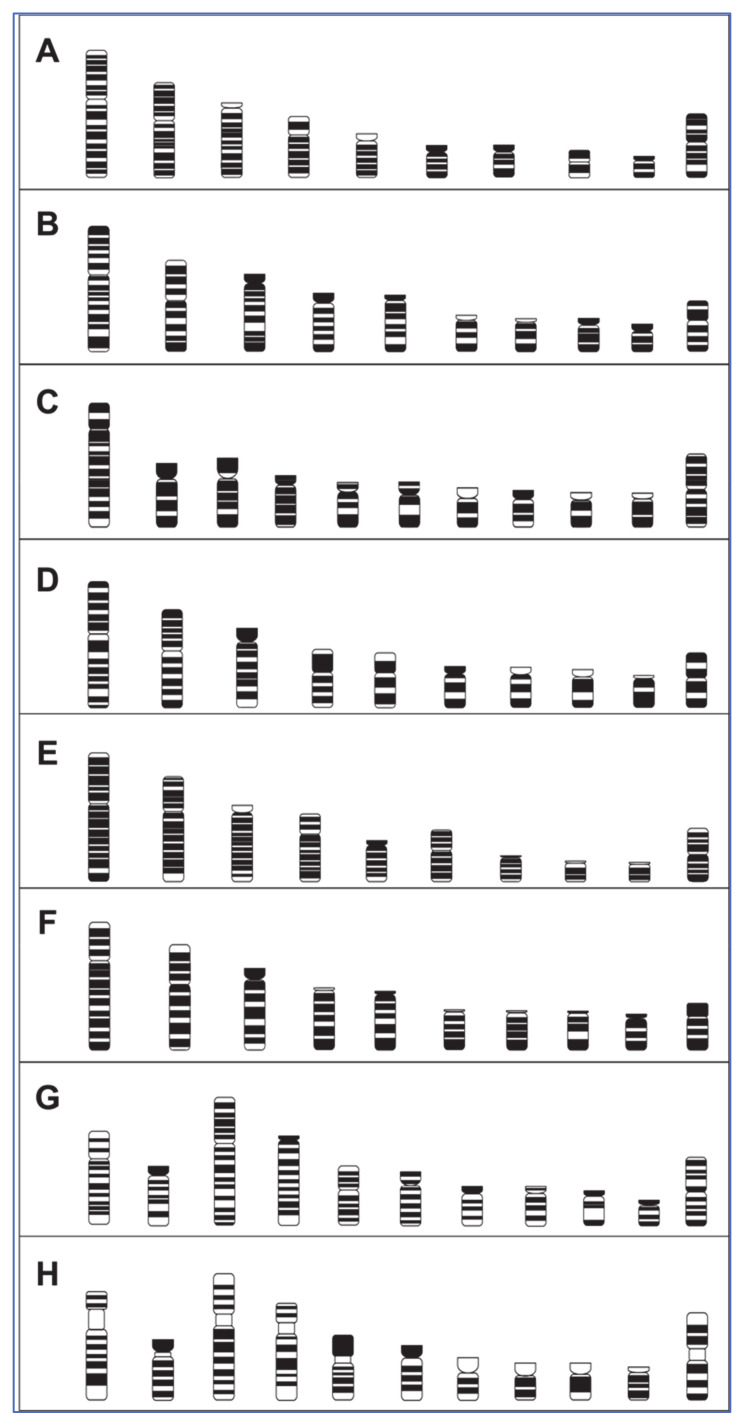
Ideograms of the macrochromosomes from all of the seven avian species compared to chicken. (**A**) chicken (*Gallus gallus*), (**B**) mallard duck (*Anas platyrhynchos*), (**C**) Eurasian woodcock (*Scolopax rusticola*), (**D**) rock dove/pigeon (Columba livia), (**E**) helmeted guinea fowl (*Numida meleagris*), (**F**) houbara bustard (*Chlamydotis undulata*), (**G**) common blackbird (*Turdus merula*), (**H**) Atlantic canary (*Serinus canaria*). Scale bar 5 μm.

**Figure 4 cells-10-00362-f004:**
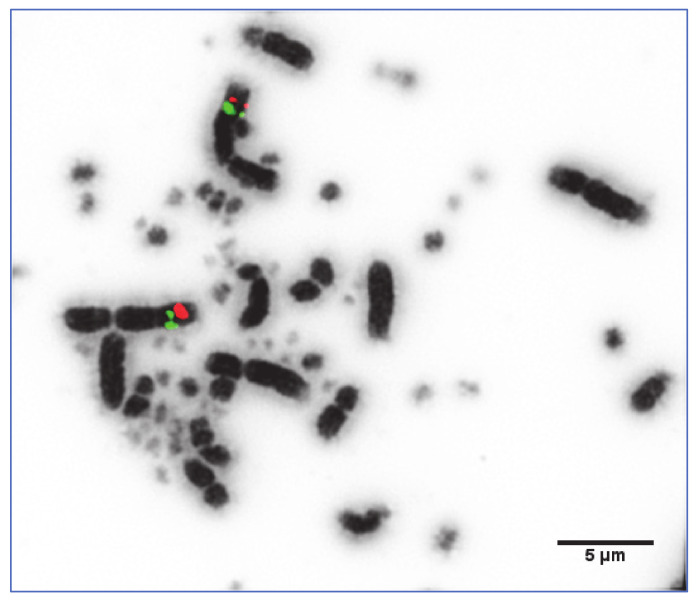
BAC clones hybridised to helmeted guinea fowl chromosome 1. The FITC (green) labelled signal represents CH261-107E2 (chicken 1 homolog), the Texas red labelled signal represents CH261-184E5 (chicken 1 homolog).

**Figure 5 cells-10-00362-f005:**
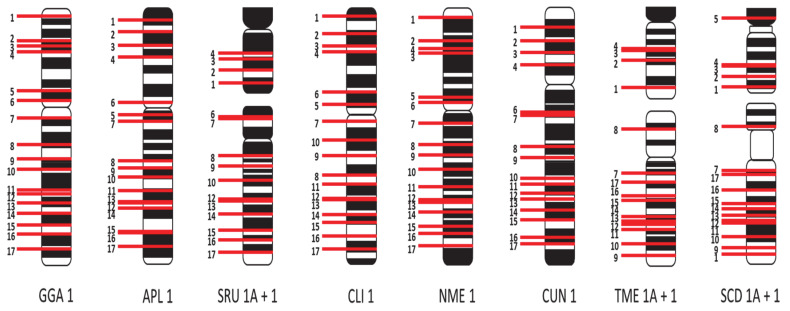
Ideograms indicating relative hybridisation positions of BACs for chicken chromosome 1, with BACs labelled 1-17 in order of position on the chicken chromosome. BAC positions are indicated for chicken (GGA) chromosome 1, mallard (APL) 1, pigeon (CLI) 1, helmeted guinea fowl (NME) 1, and houbara bustard (CUN) 1. For the common blackbird (TME), Atlantic canary (SCD), and Eurasian woodcock (SRU), BACs are indicated for chromosomes 1A (top) and 1 (bottom). The remaining chromosomal rearrangements are given in Appendix A.

**Figure 6 cells-10-00362-f006:**
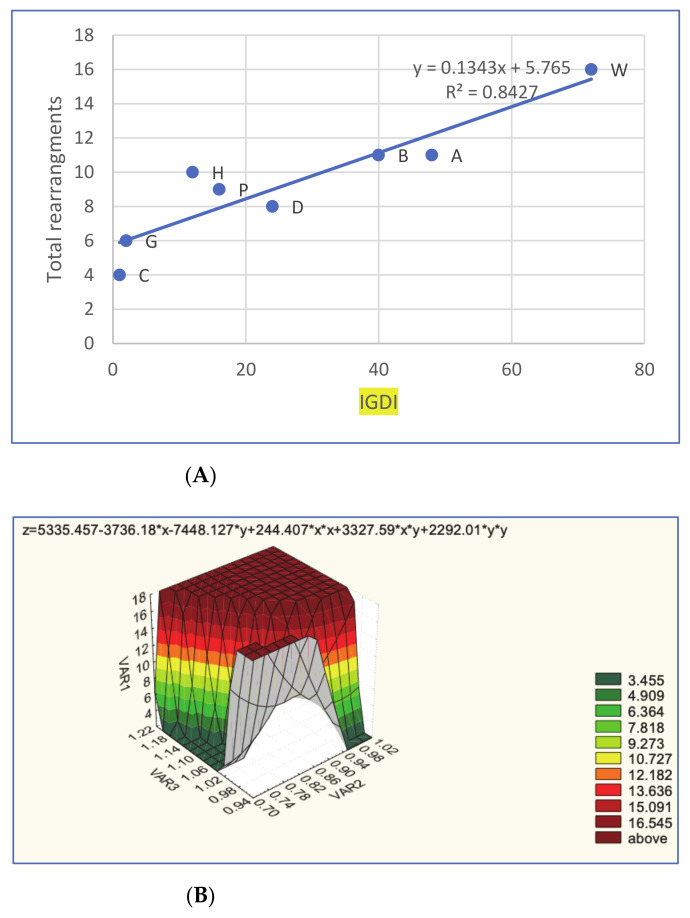
Graphical representation for correlations between rearrangement/divergence characteristics in the eight avian genomes studied. Upper (**A**): Simple linear correlation between the integrative genome/divergence index (IGDI; see details in Appendix A) and total rearrangements: C, chicken; G, helmeted guinea fowl; D, mallard duck; P, pigeon; H, houbara bustard; B, common blackbird; A, Atlantic canary; and W, Eurasian woodcock. Lower (**B**): Multiple correlation 3D surface plot (see details in Appendix A) for total number of rearrangements (VAR1), fluorescence in situ hybridisation (FISH) success rate (VAR2) and ratio of diploid number of chromosomes to the typical avian karyotype (2n/80; VAR3) across the eight avian species studied.

**Figure 7 cells-10-00362-f007:**
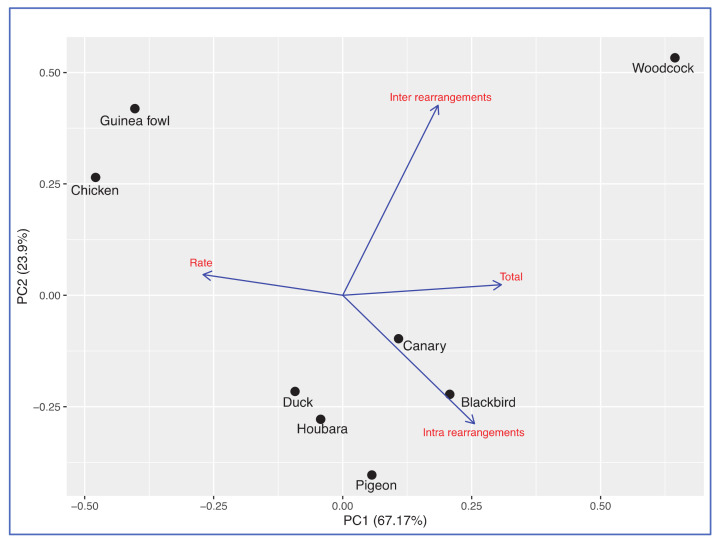
Principal Component Analysis (PCA) score plot generated for the eight bird species studied using four characteristics: BAC hybridisation success rate (Rate), and numbers of total (Total), intra- (Intra rearrangements) and interchromosomal (Inter rearrangements) rearrangements.

**Table 1 cells-10-00362-t001:** Summary of karyotype, divergence, and percentage of successful bacterial artificial chromosome (BAC) hybridisation in the eight avian species studied.

Infraclass	Order	Common Name	2n	Divergence ^1^ (Mya ^2^)	Hybridisation Success Rate (%)
Neognathae	Galliformes	chicken	78 ^3^	–	100
	Galliformes	helmeted guinea fowl	78 ^3^	47	100
	Anseriformes	duck (mallard)	80 ^4^	80	85.1
	Columbiformes	rock dove (pigeon)	80 ^5^	98	93.2
	Otidiformes	houbara bustard	76 ^6^	98	87.8
	Passeriformes	common blackbird	80 ^7^	98	78.4
	Passeriformes	Atlantic canary	80 ^8^	98	73.0
	Charadriiformes	Eurasian woodcock	96 ^6^	98	73.0
Palaeognathae	Struthioniformes	ostrich ^9^	80 ^6^	140	83.8

^1^ As estimated between the chicken and any other studied bird using TimeTree [39]. ^2^ Million years ago. 2n, diploid number of chromosomes, according to: ^3^ Shibusawa et al. [10]; ^4^ Fillon et al. [40]; ^5^ Damas et al. [13]; ^6^ O’Connor et al. [32]; ^7^Hammar [41]; ^8^ Dos Santos et al. [42]. ^9^ Ostrich is added (from O’Connor et al. [15,32]) for comparison.

**Table 2 cells-10-00362-t002:** Summary of chromosomal rearrangements occurring from the common ancestor, as determined by Genome Rearrangements in Man and Mouse (GRIMM).

Species	Inversions	Duplications	Intra-Chromosomal Translocations	Inter-Chromosomal Translocations	Fusions	Fissions	Total
Chicken	3				1		4
Guinea fowl	4				2		6
Duck	8						8
Houbara	9						9
Pigeon	11						11
Blackbird	9					1	10
Canary	4	2	2	2		1	11
Woodcock	8			3		5	16

## Data Availability

All data is contained within the manuscript and supplement, individual FISH images and karyotypes available from the authors on request.

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
