# Peer review of "Comparative Mapping of the Macrochromosomes of Eight Avian Species Provides Further Insight into Their Phylogenetic Relationships and Avian Karyotype Evolution"

_cells, 2021, doi:10.3390/cells10020362_

Round 1

Reviewer 1 Report

This study took advantage of 74 universal avian BAC probes to characterize the intrachromosomal rearrangements of bird macrochromosomes. This demonstrates that these BAC probes are useful to study chromosomal rearrangements across the bird clade. This study will be of great interests to researchers in the field of bird cytogenetics, however, the quality of writing and presentation (e.g. figures) need to be improved. Below are my comments 

L2-4, the title is exaggerating: this study does not provide insights into those birds’ phylogenetic relationships. In fact, at least for duck the phylogenetic position is incorrect. 

L37-38, a recent study characterising bird chromosomal rearrangements has been published (Liu et al. 2021 Genome research) using several chromosome-level assemblies; the authors should take this chance to discuss the advantage as well as the disadvantage of the methodology of cytogenetics vs. genomics. 

L40-41: are the authors suggesting that the FISH technique can provide a finer resolution than chromosome-level assemblies to study chromosomal changes. It’s unclear from the writing.

L156, in Figure 1, why are ‘Neognathae’ and ‘Palaeognathae’ highlighted by red circles? And why there are red underlines for ‘Neoaves’ and ‘Galloanserae’. There should be enough space to write down the species names instead of an initial letter.

L205, please point out in which species no literature about karyotype is present.

Figure 3, please label the chromosomes.

Figure 5, the font looks a bit strange.

L266, in Table 2, I believe some of the chromosomal rearrangements are shared by species. It would be good to show the chromosomal rearrangements at each internal branch.

L268-273, this is a very long sentence and the meaning is obscure. Please consider re-writing it.

L276, I cannot find the Supplementary Figure SN1a.

L276-277, If I understand it right, rearrangement is one component of the IDRI, then how can one use it to correlate with rearrangement itself - they are not two independent datasets. Should the author test the correlations between divergence and number of rearrangements. Note I don’t see the Supplementary note. 

L279-282, I don’t see the author's interpretation of this analysis. I guess the intention of this analysis is to understand how FISH success rate is correlated with the number of chromosomal rearrangements and the diploid numbers. However, FISH success rate is influenced by many factors, including species divergence. This complicates the interpretation of the multiple correlation. By looking at the correlation 3D graph, I struggle to see a major pattern. Without the authors' explanation, this analysis does not seem to be informative.  

Figure 6, the labels cannot be seen clearly because of the back background. 

L293-297, did the author generate a phylogeny based on the PCA/FAC/ALC analyses? Again I don’t see the Supplementary Note. In figure 7 one can only see the clustering of the species but it does not tell us about the phylogenetic relationship.  

L313, the fact that only 3 probes are present on the Z chromosome may underestimate the number of chromosomal rearrangements in songbirds, because the Z chromosomes are frequently to have rearrangements that are associated with speciation. Moreover, the authors counted the number of chromosomal rearrangements from the Neognathae ancestor to each species, however, we don’t know the number of rearrangements in the branch leading to each species. It’s possible that pigeon, woodcock and houbara shared some rearrangements, i.e. that rearrangement occurred in their common ancestor, while songbirds may have more rearrangements in the terminal branches.   

L368-369, the authors should point out at least one example. 

L385, duck and chicken/guinea fowl belong to Galloanserae, diverging from other neognaths (Neoaves) at least 80 millions’ years ago. Even so, the information from using the 74 BAC probes is insufficient to correctly place duck in the avian phylogeny. This strongly suggests this approach has a major limitation in phylogenetic inference.

Author Response

Comments and Suggestions for Authors

L156, in Figure 1, why are ‘Neognathae’ and ‘Palaeognathae’ highlighted by red circles? And why there are red underlines for ‘Neoaves’ and ‘Galloanserae’. There should be enough space to write down the species names instead of an initial letter.

Authors’ Response: We have revised Figure 1 following the reviewer’s suggestion as follows:

Figure 1. The MLGO input phylogenetic tree for the eight birds plus ostrich taken as an outgroup genome. The tree was visualised using the ETE v3 toolkit [37]. The respective Newick format tree can be written as (((((Atlantic canary,Common blackbird),Eurasian woodcock),(Pigeon,Houbara bustard)),((Chicken,Helmeted guinea fowl),Duck)),Ostrich); as inferred from the Prum et al. [36] phylogeny for birds. Provisional support values (1) are shown in red.

Figure 6, the labels cannot be seen clearly because of the back background.

Authors’ Response: Thanks for noticing this. We revised this figure as follows:

This better image of the graphical dependence has also been presented by us in the revised manuscript text.

L268-273, this is a very long sentence and the meaning is obscure. Please consider re-writing it.

Authors’ Response: We suggest the following revision of these lines:

Correlation analysis for the eight avian species (see Supplementary Note 1) revealed an association between chromosomal rearrangement patterns, on the one hand, and overall karyotype/genome organisation and divergence time, on the other. PCA/FAC/ALC-based assessments (see Supplementary Note 2) provided further information in support of the cytogenetically observed specifics of chromosome changes within and between individual lineages in the infraclass Neognathae.

L276, I cannot find the Supplementary Figure SN1a.

Authors’ Response: This figure is part of the Supplementary Note 1. We are sorry for not attaching the Supplementary Notes 1 and 2 at the time the original submission. We provide them now.

L276-277, If I understand it right, rearrangement is one component of the IDRI, then how can one use it to correlate with rearrangement itself - they are not two independent datasets. Should the author test the correlations between divergence and number of rearrangements. Note I don’t see the Supplementary note.

Authors’ Response: No, this integrative index did not include rearrangement as one of its three components. We did make an error in designating it and explaining it in the Supplementary Note 1. The attached revised manuscript and Supplementary Note 1 now reflect the appropriate corrections.

L279-282, I don’t see the author's interpretation of this analysis. I guess the intention of this analysis is to understand how FISH success rate is correlated with the number of chromosomal rearrangements and the diploid numbers. However, FISH success rate is influenced by many factors, including species divergence. This complicates the interpretation of the multiple correlation. By looking at the correlation 3D graph, I struggle to see a major pattern. Without the authors' explanation, this analysis does not seem to be informative.

Authors’ Response: Our attempts to carry out this analysis were of an auxiliary and applied nature, and therefore, we did not try to provide a deeper discussion on this topic in the text. Here we simply estimated the number of chromosomal rearrangements depending on two parameters, success rate and 2n/80, that can be used to characterise genome/karyotype similarity (or dissimilarity) between species. In this multiple correlation estimation, the number of chromosomal rearrangements was a main estimated parameter (VAR1) that definitely correlated with two others, success rate (VAR2) and 2n/80 (VAR3). The use of these parameters in the calculations showed a fairly high agreement with the experimental data, which makes it possible to judge the prospects of their use in conducting such analytical studies.

L293-297, did the author generate a phylogeny based on the PCA/FAC/ALC analyses? Again I don’t see the Supplementary Note. In figure 7 one can only see the clustering of the species but it does not tell us about the phylogenetic relationship.

Authors’ Response: We produced several clustering patterns and compared them to the known phylogeny. We do apologise for not attaching Supplementary Note 2 to the original submission and provide it now. For example, using ALC clustering (as shown in Supplementary Note 2), we plotted a sort of cladograms and compared them to the known phylogeny. Altogether, these PCA/FAC/ALC clustering patterns are simple and obvious showing that they were in agreement with the known phylogeny, except for the duck as was clearly stated in Lines 293-297 of the originally submitted manuscript.

Reviewer 2 Report

The authors report a comparative chromosome analysis in eight avian species. Using a set of 74 chicken BAC probes, they assessed the chromosome rearrangement patterns and phylogenetic relationships of the analysed species. The manuscript is concise, well written, and based on a large amount of experimental data presented as Supplementary materials. I recommend the manuscript for publishing after a minor revision:

Specific Comments:

  1. p. 3, lines 118-119: “…Post-hybridization washes were 30 s in 2× SSC/0.05% Tween 20 at room temperature prior to counterstaining…” Can you check, please? I would expect using a stringent washing procedure
  2. p. 9, lines 268, 271: I could not find any Supplementary Note 1 or 2. Any list of Supplementary materials is also missing at the end of the paper (following Conclusions)
  3. p. 9, lines 274-282: Nice correlation, can you also add P values?

Author Response

Comments and Suggestions for Authors

  1. 9, lines 268, 271: I could not find any Supplementary Note 1 or 2. Any list of Supplementary materials is also missing at the end of the paper (following Conclusions)

Authors’ Response: We apologise for not attaching Supplementary Notes 1 and 2 at the time of submitting the manuscript. This is our fault. We are attaching these supplementary notes to the revision and provided their list at the end of the paper as follows:

Supplementary Materials: The following are available online at https://www.mdpi.com/...: Supplementary Note 1: Correlation analyses; and Supplementary Note 2: PCA, FAC and ALC clustering analyses.

  1. 9, lines 274-282: Nice correlation, can you also add P values?

Authors’ Response: We calculated p-values and they were significant and equal to 0.017462, 0.0013 and 0.000012, respectively. These appropriate p-values were added in the revised article.
